# Large deep-sea zooplankton biomass mirrors primary production in the global ocean

S. Hernández-León [1✉], R. Koppelmann [2], E. Fraile-Nuez[3], A. Bode [4], C. Mompeán[4], X. Irigoien[5,6], M. P. Olivar[7], F. Echevarría [8], M. L. Fernández de Puelles[9], J. I. González-Gordillo [8], A. Cózar[8], J. L. Acuña [10], S. Agustí [11] & C. M. Duarte [11,12]

The biological pump transports organic carbon produced by photosynthesis to the meso- and bathypelagic zones, the latter removing carbon from exchanging with the atmosphere over centennial time scales. Organisms living in both zones are supported by a passive flux of particles, and carbon transported to the deep-sea through vertical zooplankton migrations. Here we report globally-coherent positive relationships between zooplankton biomass in the epi-, meso-, and bathypelagic layers and average net primary production (NPP). We do so based on a global assessment of available deep-sea zooplankton biomass data and large-scale estimates of average NPP. The relationships obtained imply that increased NPP leads to enhanced transference of organic carbon to the deep ocean. Estimated remineralization from respiration rates by deep-sea zooplankton requires a minimum supply of 0.44 Pg C $y^{-1}$ transported into the bathypelagic ocean, comparable to the passive carbon sequestration. We suggest that the global coupling between NPP and bathypelagic zooplankton biomass must be also supported by an active transport mechanism associated to vertical zooplankton migration.

[1] Instituto de Oceanografía y Cambio Global, IOCAG, Universidad de Las Palmas de Gran Canaria, Unidad Asociada ULPGC-CSIC, Campus de Taliarte, 35214 Telde, Gran Canaria, Canary Islands, Spain. [2] Institut für Marine Ökosystem- und Fischereiwissenschaft, Universität Hamburg, Grosse Elbstrasse 133, Hamburg, Germany. [3] Instituto Español de Oceanografía, Vía Espaldón, Dársena Pesquera, 38180 Santa Cruz de Tenerife, Canary Islands, Spain. [4] Instituto Español de Oceanografía (IEO), Centro Oceanográfico de A Coruña, 15080 A Coruña, Spain. [5] AZTI, Herrera Kaia, Portualdea z/g, Pasaia, Gipuzkoa 20110, Spain. [6] Ikerbasque, Basque Foundation for Science, Bilbao, Spain. [7] Institut de Ciències del Mar, CSIC, 08003-Barcelona, Catalunya, Spain. [8] Instituto Universitario de Investigación Marina (INMAR), Universidad de Cádiz, 11510 Puerto Real, Cádiz, Spain. [9] Instituto Español de Oceanografía (IEO), Centro Oceanográfico de Baleares, Muelle de Poniente s/n, 07015 Palma, Spain. [10] Observatorio Marino de Asturias, Departamento de Biología de Organismos y Sistemas, Universidad de Oviedo, Oviedo, Spain. [11] Red Sea Research Center (RSRC), King Abdullah University of Science and Technology (KAUST), Thuwal 23955, Saudi Arabia. [12] Computational Bioscience Research Center (CBRC), King Abdullah University of Science and Technology (KAUST), Thuwal 23955, Saudi Arabia. ✉email: shernandezleon@ulpgc.es

Oceanic primary production sets constraints to the flow of organic carbon into the deep sea and, therefore, the supply of carbon to support the food web of the dark ocean[1], which, with 99% of the ocean volume, is the largest habitat on Earth[2]. Carbon export and sequestration through passive flux decreases sharply with depth (Martin curve[3]) due to consumption by zooplankton and micronekton, and prokaryote remineralization. This implies that only a small percentage of the gravitational flux leaving the upper ocean reaches the bathypelagic layer[4–6]. Because of the exponential decreasing nature of the sinking flux with depth, zooplankton biomass (ZB) in the deep sea has been assumed to be so low as to be negligible relative to all ocean biomass, as stated in a recent summary[7].

However, recent findings for the subtropical and tropical oligotrophic ocean challenge traditional views and point at a much more efficient delivery of carbon from primary production in the photic layer to the dark ocean. In particular, (1) mesopelagic fish biomass is, at least, one order of magnitude larger than hitherto assumed, increasing with increasing primary production in the overlaying waters[8]. In this sense, it has been observed the vertically migrating fauna supports a large active carbon flux into the mesopelagic layer, particularly in productive areas[9]; (2) healthy photosynthetic cells are widespread in the dark ocean, pointing at a faster, and therefore more efficient delivery of carbon to the bathypelagic layer than assumed[10]; and (3) assessments of the fraction of carbon exported from the photic layer reaching the deep sea at 2000 m depth yield a much greater estimate (reaching 20%) than often assumed[11]. One implication of these various lines of evidence is that ZB in the deep sea could be much higher, and more closely coupled to primary production in the epipelagic layer than until now assumed. This suggestion is supported by a recent assessment that ZB in the bathypelagic zone of the subtropical and tropical Atlantic Ocean is half of that in the epipelagic layer and twice that in the mesopelagic layer, and correlates

with chlorophyll $a$ concentration in the overlaying epipelagic layer[12].

Here we report the global distribution of ZB at epi-, meso-, and bathypelagic depths based on data acquired during the Malaspina Circumnavigation Expedition[8], which surveyed ZB in subtropical and tropical oceans during 2010 and 2011, amended with published estimates of deep-sea ZB (see "Methods"). We then examine the relationship between average net primary production (NPP) and ZB estimated during the survey, and provide a first-order estimate of the global contribution of zooplankton to carbon cycling in the deep sea. Additional support for the role of zooplankton metabolism at depth was provided by the analysis of nitrogen stable isotope composition of zooplankton and estimations of respiration from enzymatic assays in Malaspina samples (see "Methods"). We found independent and globally coherent positive relationships between ZB in the meso-, and bathypelagic layers of the ocean and average NPP in the overlying photic layer. The relationships obtained imply that increased primary productivity in the euphotic layer leads to enhanced transference of organic carbon to the deep ocean. Estimated remineralization from respiration rates by deep-sea zooplankton requires a minimum supply of $0.44\,\mathrm{Pg\,C\,y^{-1}}$ transported into the bathypelagic ocean, comparable to the magnitude of passive carbon sequestration[5,6], which itself is required to support deep-sea microbial carbon demands. Thus, we suggest that the global coupling between primary production and bathypelagic ZB must be supported by passive carbon supply, including episodic events of high carbon delivery, as well as an important contribution from an active transport mechanism associated to vertical zooplankton migration.

## Results

**Relationships between deep-sea ZB and primary production.** ZB attenuated with depth from the upper to the bathypelagic ocean at an average (±SE) rate of $1.21 \pm 0.04\,\mathrm{km^{-1}}$ (Fig. 1 and Supplementary Table 1). Data gathered showed significant, positive relationships between average NPP and ZB along the tropical and subtropical ocean sampled along the Malaspina Expedition in the epi- ($\mathrm{LnNPP} = 0.690 + 0.915\,\mathrm{LnZB}$, $r^2 = 0.41$, $p < 0.001$, $n = 42$), meso- ($\mathrm{LnNPP} = -3.639 + 1.647\,\mathrm{LnZB}$, $r^2 = 0.36$, $p < 0.001$, $n = 43$), and bathypelagic (1000–2000 m in this data set, $\mathrm{LnNPP} = -7.445 + 2.064\,\mathrm{LnZB}$, $r^2 = 0.61$, $p < 0.001$, $n = 35$) layers (Fig. 2). Strikingly, the relationships converged showing similar biomass in all three layers at the highest NPP (Fig. 2). Independent estimates derived from acoustic backscatter intensity acquired down to 4000 m depth using a Lowered Acoustic Doppler Current Profiler (LADCP) along the Malaspina Expedition (Fig. 3), a proxy for zooplankton and micronekton biomass[13], confirmed the existence of significant positive relationships between acoustic backscatter intensity and average NPP in the mesopelagic (200–1000 m, $r^2 = 0.065$, $p < 0.05$, $n = 108$), and the upper (1000–2000 m, $r^2 = 0.61$, $p < 0.001$, $n = 107$), intermediate (2000–3000 m, $r^2 = 0.61$, $p < 0.001$, $n = 101$), and lower (3000–4000 m, $r^2 = 0.54$, $p < 0.001$, $n = 90$) bathypelagic layers, also converging in similar values for all three layers at high NPP (Supplementary Fig. 1). An analysis of independent, published estimates of primary production[14] and the global data set of ZB assembled here further validated the results obtained in the Malaspina Expedition. In particular, the average ZB in the upper bathypelagic zone increased with increasing average NPP of the Longhurst's biogeochemical ocean provinces where ZB was sampled[14] ($r^2 = 0.35$, $p < 0.001$, $n = 31$ for all data, except the Red Sea, cf. "Methods"), with the correlation being particularly stronger for the Trades, Westerlies, and Polar domains ($r^2 = 0.41$, $p < 0.001$, $n = 24$, Supplementary Fig. 2 and see "Methods").

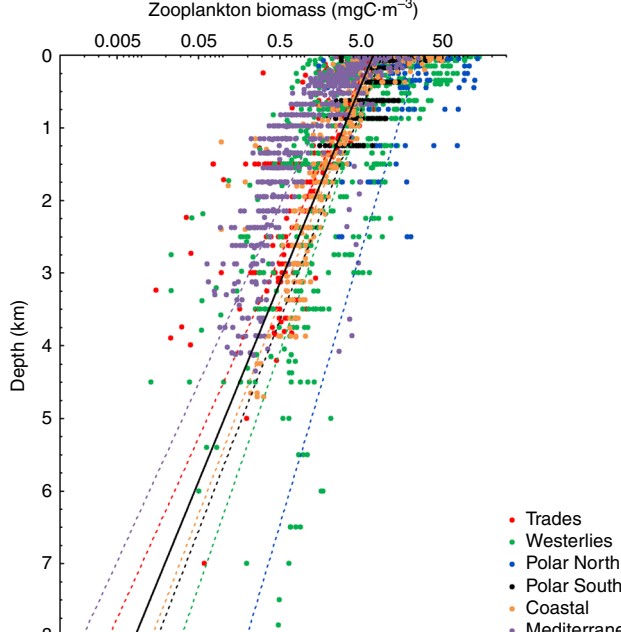

**Fig. 1 Vertical distribution of zooplankton biomass.** Vertical profiles of zooplankton biomass in different oceanic biomes as defined by Longhurst et al.[14]. Dotted lines represent the fitted regression equations shown in the Supplementary Table 1. Black line is the general equation for all the data, also given in the Supplementary Table 1. Note that the X-axis is in logarithmic scale.

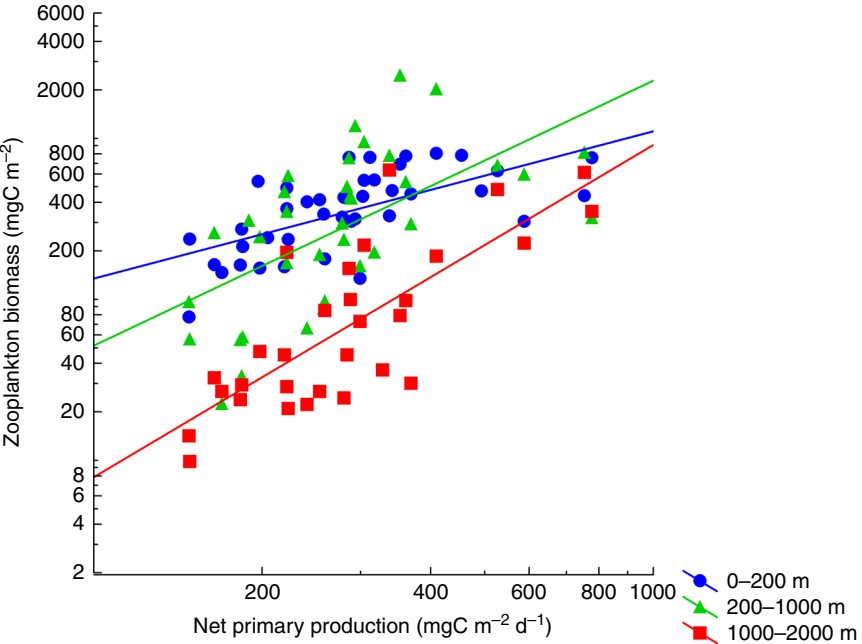

**Fig. 2 Zooplankton biomass and primary production.** Relationships between net primary production and zooplankton biomass in the epi- (0–200 m), meso- (200–1000 m), and bathypelagic (1000–2000 m) layers during the Malaspina Circumnavigation Expedition. Solid lines represented the fitted regressions. Note that both axis are in logarithmic scale.

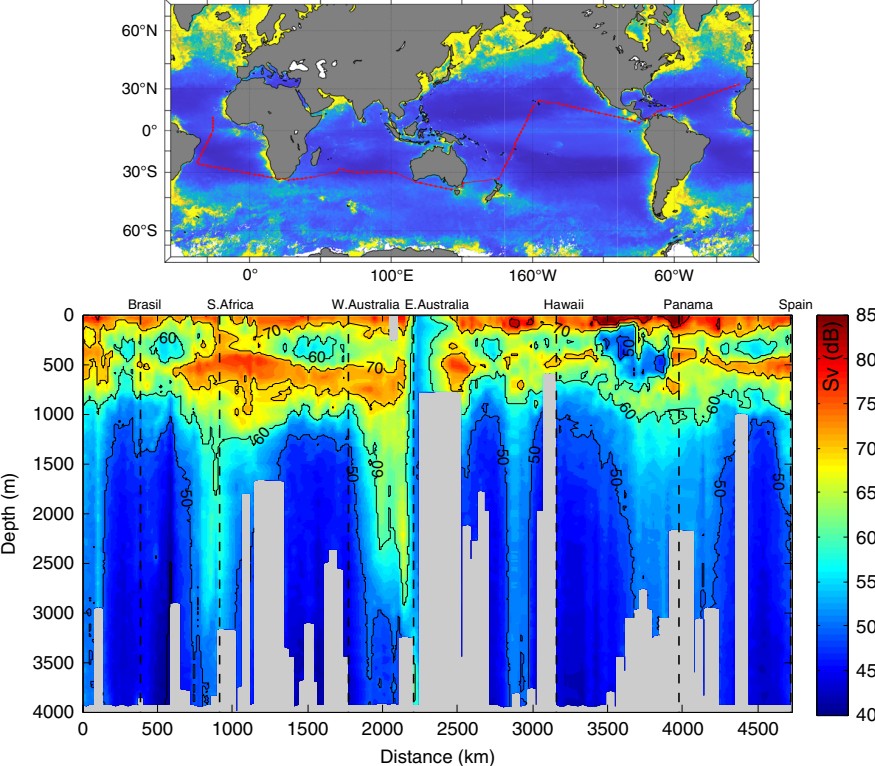

**Fig. 3 Acoustic backscatter.** Backscatter volume strength (SV) in dB obtained during the Malaspina Circumnavigation Expedition using the Lowered Acoustic Doppler Current Profiler (LADCP) attached to the rosette sampler. Red line in the upper panel shows the Circumnavigation journey. Observe the deep scattering layer (DSL) all along the transect around 500 m depth, the almost absence of DSL in the oxygen minimum zone of the Pacific Ocean, and the global increase in backscatter at bathypelagic depths below areas of higher productivity approaching the continents (vertical dashed lines) and equatorial upwelling zones. The distance in the *x*-axis are the km from the beginning to the end of the transect.

**Stable isotopes vertical distribution.** Correlations between average NPP and ZB at depth could be driven by the sinking of particulate matter fueling zooplankton and micronekton in the deep sea[15], by zooplankton[16,17] migrating vertically to feed upon organisms in shallower layers, or both. In addition to the larger than expected biomass of zooplankton at depth, there is evidence for [15]N enrichment implying a higher trophic level of deep-sea zooplankton compared to that in surface waters, as heavier isotopes are progressively accumulated in upper trophic levels[18]. A general increase in nitrogen isotopic signature of zooplankton with depth has been widely observed[19]. Samples collected along the Malaspina Expedition also showed the average isotopic nitrogen signature of zooplankton to be progressively enriched in heavier isotopes from the surface to 3000 m (Supplementary Fig. 3 and Supplementary Table 2). This enrichment was significant in the Trades biome provinces but not in provinces of the Coastal biome, thus suggesting the existence of regional differences in the links between zooplankton and surface productivity. Maximum enrichment at depth could be expected when the food web largely depends on repeatedly recycled nitrogen, as found in low productivity regimes[20]. In contrast, seasonally productive ecosystems display large variations in plankton isotopic signatures that are attributed to changes in the amount of organic matter channeled through microbial pathways, particularly after the peaks in primary production[20,21]. In addition to in situ feeders, bathypelagic zooplankton communities include seasonal ontogenetic migrants overwintering at depth and not feeding there[22], and organisms using the deep sea as a refuge from predators[16]. Hence, the potentially higher episodic sedimentation[10] and the stronger active flux performed by zooplankton and micronekton[9,23] should promote the homogenization of stable isotopes in the water column in productive areas (see Supplementary Fig. 3). This observation, therefore, supports a faster downward energy flux, fueling ZB in productive areas.

**Deep-sea ZB and remineralization.** Our results show that ZB in the twilight zone and deep sea are much larger than hitherto assumed and that ZB in the dark ocean increase with increasing average primary production in epipelagic waters. We estimated a total oceanic ZB of 1.4 Pg C, with 34%, 47%, and 19% of this biomass located in the epi-, meso-, and bathypelagic layers, respectively. Supporting this high biomass and the associated carbon demands requires a much larger carbon supply to the dark ocean (see Fig. 4) than previously estimated[4–6]. The respiratory demands to support the mesozooplankton biomass in the dark ocean provides a minimum estimate of the carbon inputs required. We provided a first-order approximation of the magnitude of carbon remineralization by bathypelagic zooplankton derived from estimates of the enzymatic activity of the electron transfer system (ETS) conducted during the Malaspina Expedition ("Methods"). These estimates ranged from 0.01 to 1.97 $\mu$l $O_2$ mg$^{-1}$ dry weight h$^{-1}$ (average 0.33 $\mu$l $O_2$ mg$^{-1}$ dry weight h$^{-1}$, Supplementary Table 3) corresponding to carbon turnover rates of ZB of 0.001 to 0.062 d$^{-1}$ (mean ± SD, 0.010 ± 0.012 d$^{-1}$, $n$ = 33). This is slightly higher than mean values of 0.006 ± 0.003 d$^{-1}$ for both upper (1000–2000 m depth, $n$ = 57) and lower (2000–3000 m depth, $n$ = 47) bathypelagic zones reported in the past[24], but slightly lower than those derived using the global-bathymetric model of Ikeda[25], which yield an average value of bathypelagic zooplankton respiration of 0.56 ± 0.10 $\mu$l $O_2$ mg$^{-1}$ dry weight h$^{-1}$, corresponding to a carbon turnover of ZB of 0.018 ± 0.003 d$^{-1}$. The estimated respiratory demands does not include zooplankton excretion rates, which would increase the carbon demands of zooplankton by about 20%, as observed in the mesopelagic zone[26]. Thus, the estimate of turnover of deep-sea

zooplankton carbon obtained from ETS activity of about 0.01 d$^{-1}$ provides an estimate of the carbon supply to deep-sea zooplankton amounting, when coupled to their estimated global biomass integrated across Longhurst oceanic biogeochemical provinces, to 0.44 Pg C y$^{-1}$ remineralized by bathypelagic zooplankton (0.28 Pg C y$^{-1}$ within the 1000–2000 m layer, 0.11 Pg C y$^{-1}$ for the 2000–3000 m layer, and 0.05 Pg C y$^{-1}$ for the 3000–4000 m layer, Table 1 and Supplementary Table 4). This assessment is highly conservative, as we used moderate respiration rates and we did not include the contribution of excretion and demands from macrozooplankton and micronekton.

**Discussion**

Estimates of passive carbon sequestration into the deep ocean derived with the same up-scaling criteria of biomes and provinces[14] resulted in 0.72 Pg C y$^{-1}$ at the base of the pycnocline and 0.33 Pg C y$^{-1}$ at 2000 m depth[6], similar to other global assessments[5,27,28]. Hence, our estimate of the carbon remineralized by zooplankton is consistent with that of the passive flux of sedimentary organic carbon into the bathypelagic layer. Provided that bacterial respiration in the bathypelagic layer[29] is estimated to be in the order[30] of 1.3–1.6 Pg C y$^{-1}$, with upper estimates[29] up to 18.0–20.4 Pg C y$^{-1}$, there is no scope for both bathypelagic prokaryotes and zooplankton being supported by the carbon supplied through passive sinking alone[29–31] (see Fig. 4).

Carbon export from the photic layer includes a passive[5,6] and active flux[32–34], as well as an export of dissolved organic carbon (DOC) (Fig. 4), the latter estimated to account for ~20% of global passive export production but about half of the total passive organic carbon export in the oligotrophic subtropical oceans[35]. However, DOC export does not appear to penetrate beyond the mesopelagic layer, where it supports on average only 8.4% of respiratory demands[36]. Hence, neither passive POC nor DOC fluxes can account for the high carbon demand by mesopelagic and bathypelagic zooplankton and prokaryotes.

Diel vertical movements of zooplankton between the epipelagic and the bathypelagic zones were described long ago[16] and vertical migrations from the surface to the deep sea were also reported for micronekton[37,38], along with recent reports using moored ADCPs demonstrating vertical migrations through the upper bathypelagic zone[39–41]. Thus, the active carbon shunt by macroplankton[16,33] and micronekton[37,38] by feeding in upper layers and migrating down to deeper waters is likely to be large, since similar values of zooplankton and micronekton active fluxes were observed in the mesopelagic layer[9,42,43]. Yet, the magnitude of the carbon transport to the bathypelagic zone associated with these migrations remained ignored. As in the mesopelagic zone, respiration by prokaryotes and zooplankton in the bathypelagic zone is much higher than can be supported by the passive flux. Thus, an additional contribution from active carbon flux is required to balance the budget (see Fig. 4).

The active carbon transport to depth performed by the pelagic fauna involves two mechanisms operating at different nested time scales: diel vertical migration[26,32,33] and ontogenetic vertical migration[22,44]. These mechanisms were hypothesized to operate in tandem through the so-called Vinogradov's ladder of migration[16] where animals feeding in the upper, productive ocean and migrating subsequently into deeper layers provide resources and/or serve as food for meso- and bathypelagic populations feeding in the upper levels of their depth range and migrating to deeper waters. Whereas zooplankton and micronekton can migrate daily vertical distances of about 400–500 m, passive sinking would require, at a sinking rate of 10–100 m d$^{-1}$ for POC[15], 4–50 days. As fragmentation of particles seemed to control sequestration of organic carbon as recently observed[15,45], the role of small particles

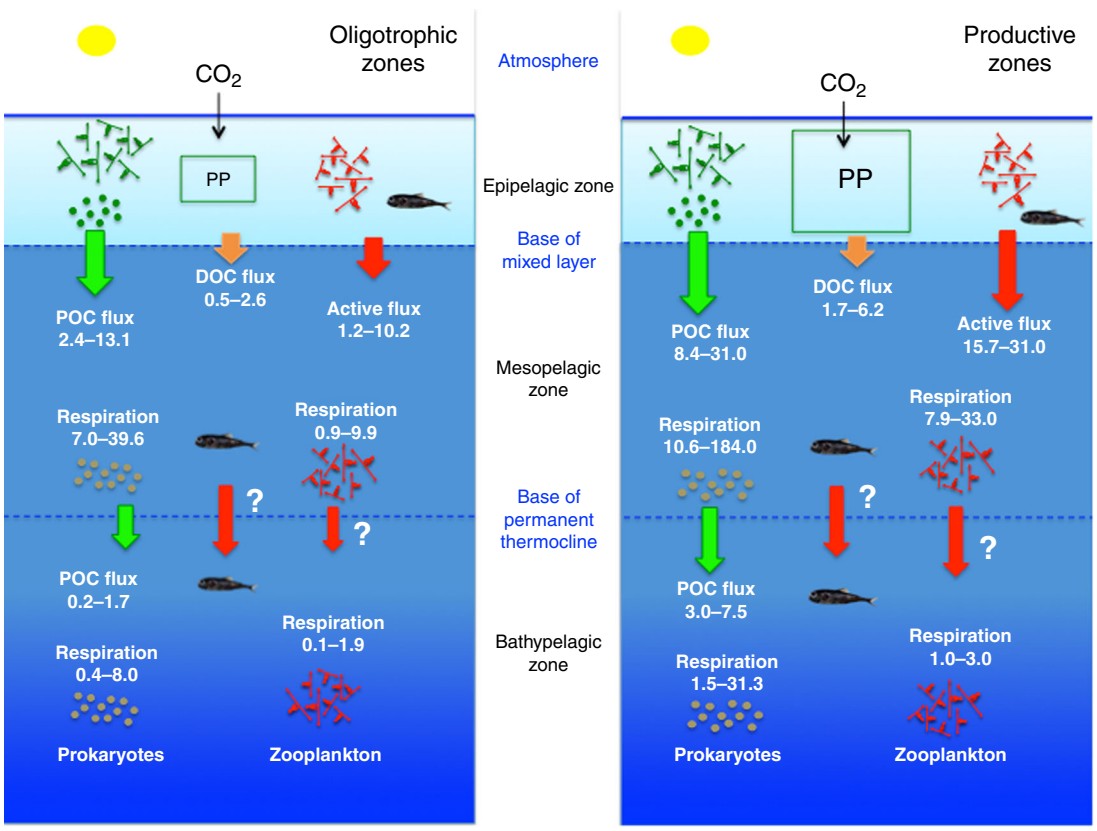

**Fig. 4 Carbon export and sequestration.** Synthesis of carbon export values (from the epipelagic towards the mesopelagic zone) and sequestration (from the mesopelagic towards the bathypelagic zone) of particulate organic carbon (POC, green arrows) flux, active flux due to migrant zooplankton, and micronekton (red arrows), and estimated dissolved organic carbon (DOC, orange arrows) flux (as 20% of POC flux, see text). Values are given in gC m$^{-2}$ y$^{-1}$, to compare values in oligotrophic (left panel) and productive systems (right panel). POC and active fluxes are higher in productive zones, as expected, but active export flux is proportionally higher in productive zones as recently observed in the tropical and subtropical Atlantic Ocean[9]. Values of prokaryote and zooplankton respiration are also higher than POC, DOC, and active fluxes for both export and sequestration. Active sequestration flux is, at present, unknown (reflected by a question mark in both panels) and it should also explain, at least in part, the higher respiration rates in the bathypelagic zones. Finally, active sequestration flux and respiration by macrozooplankton and micronekton should tend to balance the budget (see text), and they are also represented as question marks in the Figure.

**Table 1 Estimates of zooplankton remineralization in the three bathypelagic layers assessed.**

| Depth interval and domain | Remineralization 1000-2000 m, gC m$^{-2}$ y$^{-1}$ | n | Area of provinces 10$^6$ km$^2$ Longhurst et al.[14] | Remineralization TgC y$^{-1}$ | Remineralization scaled Pg C y$^{-1}$ |
|---|---|---|---|---|---|
| **1000-2000 m** | 1.91 | 274 | 265.7 (81%) | **225.7** | **0.28** |
| Trades | 0.35 | 80 | 130.1 | 43.2 | |
| Westerlies | 3.00 | 125 | 107.0 | 119.1 | |
| Polar | 4.62 | 39 | 15.2 | 48.2 | |
| Coastal | 1.03 | 30 | 13.4 | 15.2 | |
| **2000-3000 m** | 0.89 | 113 | 192.59 (59%) | **62.8** | **0.11** |
| **3000-4000 m** | 0.16 | 66 | 114.36 (35%) | **16.3** | **0.05** |
| **PgC y$^{-1}$** | | | | | **0.44** |

Percentage in brackets below of "Area of provinces" are the portions of the ocean accounted for in our review. Last column gives the total remineralization values scaled to the ocean surface given by Longhurst et al.[14]. Remineralization values for each bathypelagic layer are given in bold.

and, therefore, the slower sinking rates should be the rule[15]. Hence, active C transport along the Vinogradov's ladder is, at least, 4- to 50-fold faster than gravitational transport of POC. Vinogradov's ladder of migration also implies an increase in trophic level with depth, as zooplankton in one layer serve as prey to zooplankton in the layer below[17]. The observed increase in trophic level, inferred from increasingly higher nitrogen isotopic composition of ZB with depth is, therefore, consistent with Vinogradov's ladder of migration. Carbon transport along the Vinogradov's ladder of migration[16] bypasses microbial remineralization with the potential

to accelerate transport through depth, compared to passive sinking, and greatly increases the carbon flux to the deep sea. This is of consequence, as part of the organic carbon reaching the deep sea is effectively sequestered over time scales relevant to climate change.

The magnitude of the carbon transport is related to ZB at the different layers and should ultimately be related to productivity in the euphotic zone supporting the flux. However, a relationship between primary production and ZB at depth had not been demonstrated to date, despite more than half a century of research efforts. Here we demonstrate that high primary production propagates into high ZB at depth, involving one order of magnitude increase in bathypelagic ZB from oligotrophic (ca. 200 mgC m$^{-2}$ d$^{-1}$) to mesotrophic ocean layers (ca. 800 mgC m$^{-2}$ d$^{-1}$), implying an order of magnitude increase in remineralization in the deep sea. This is in accordance to recent results showing that ZB at depth in the subtropical and tropical Atlantic Ocean increased with surface chlorophyll $a$[12], which is related to primary production. However, a large variability was found in the epipelagic layer (0–200 m depth, Fig. 2 and Supplementary Fig. 1) in relation to average primary production because of the mismatch of measuring biomass or backscatter at a given time during the cruise and the remote-sensing average used for primary production in a rather highly dynamic system such as the epipelagic zone. In addition, horizontal water mass transport rates are five- to ten-fold faster in the epipelagic than the meso- and bathypelagic[46], also leading to higher variability in the relationship between ZB and average NPP in the epipelagic layer. The highest primary production was observed in the coastal domain (mainly close to upwelling zones) but these high values were not coupled to high backscatter in the epipelagic zone as upwelling zones export particles and ZB horizontally[47]. In contrast, however, ZB was better correlated with average primary production in the bathypelagic zone, as the biomass there is the result of long-term primary production in the upper layers.

A relationship between primary production and active flux by both zooplankton and micronekton was reported at the basin scale in the Atlantic Ocean[9], also showing a much higher slope than the relationship with passive flux. Thus, the higher active flux observed in the mesopelagic zone in productive areas[9] should also be found in the bathypelagic zone as active flux is highly related to biomass[9]. The coupling between primary production and bathypelagic ZB demonstrated here requires an active, rather than passive, carbon transport mechanism. This process relays carbon to the deep-sea shunting the much slower passive flux resolved with sediment traps and other approaches[5,6,15], and results in a minimum of 0.44 Pg C y$^{-1}$ supplied by zooplankton into the bathypelagic ocean.

Current estimates of the biological pump transferring carbon to the dark ocean embedded in depictions of the global carbon cycle[48] consider passive fluxes alone (i.e., POC and DOC fluxes in Fig. 4). Our estimates that the active organic carbon export from the photic layer must be of a similar magnitude as passive POC + DOC flux to satisfy calculated carbon demands implies that the biological pump exports twice as much carbon to the twilight and dark ocean as included in current depictions of the global carbon budget[48]. Moreover, whereas most of the DOC flux is remineralized within the mesopelagic layer[36], the Vinogradov ladder connects predator–prey active transport chains acting in tandem to deliver a minimum of 0.44 Pg C y$^{-1}$ into the bathypelagic ocean, where carbon is sequestered over time scales relevant to climate projections.

Hence, we conclude that the large magnitude of deep-sea ZB and the conservative estimate of the associated carbon transport into the bathypelagic ocean provided here point at a need to reconsider the oceanic carbon budget to account for the role of pelagic fauna as core components of the biological pump, helping to reconcile estimates of carbon supply with those of demand derived from respiratory organic carbon demands by different components of the deep-sea ecosystems[29–31].

## Methods

**Zooplankton biomass.** We reviewed 274 profiles (Supplementary Fig. 4) of ZB < 5 mm when this information was available but always excluding macrozooplankton and micronekton. Due to the different mesh sizes (200, 300, or 500 μm) used by the different authors in our review, the obtained values are in some cases an underestimation of mesozooplankton biomass for the smaller size classes. In addition to this, the inherent subsampling problems using nets, such as escapement of organisms, underestimates our biomass values. Data not available in tables (marked with an asterisk in the Supplementary Table 5) were obtained from data points of figures using GraphClick software v3.0. Biomass in different units were converted to carbon using different published conversion factors[49]. Values from different depth layers were centered at the mean depth, averaged in each profile, and expressed as standing stock below one square meter. Median values were calculated in order to obtain the carbon flux for each province (sensu Longhurst et al.[14]). The Red Sea was excluded from our analysis as it was a clear outlier showing high NPP and quite low biomass in the anomalously warm (21°C at 1000 m depth) and oxygen-depleted (<0.2 mLO$_2$ L$^{-1}$) bathypelagic waters[50]. Data from bathypelagic zones of Coastal Domains normally showed lower biomass in comparison with NPP as the high values of the latter are restricted to the coastal zone, while deep-sea zooplankton hauls were obtained far from the coast. Organisms captured during the Malaspina circumnavigation (13.5% of the data) were collected using a 0.5 m$^2$ Multinet Sampler (Hydrobios) equipped with 5 nets of 300 μm mesh and a flowmeter to measure the volume of water filtered. Stratified tows were performed by day in vertical hauls from 3000 m to the surface. Samples were then fixed on board and stored in a 4% buffered formalin and seawater for further analysis. In the laboratory, samples were scanned using an Epson Perfection 4990 PHOTO scanner at a resolution of 1200 dpi, processed using the software ZooImage 1 (version 1.2-1) to measure abundance. Biomass was obtained converting the area of each individual from different taxonomic groups to dry weight[51].

**Stable isotopes.** Natural abundance of stable nitrogen isotopes was expressed as δ$^{15}$N (‰) relative to atmospheric nitrogen and determined in zooplankton specimens collected from subsamples of the different water layers. Several individuals were combined to obtain at least 10 μg N per sample. In most cases these combined samples included several species or genera of the same trophic category according to the literature[52]. Isotopic determinations were made with a mass spectrometer (Finnigan Mat Delta Plus) coupled to an elemental analyzer (Carlo-Erba CHNSO 1108). Further details of the isotopic determinations can be found elsewhere[53]. In this study, we report vertical profiles of natural abundance averaged over all zooplankton specimens analyzed during the cruise. Stable isotope data were used here as an index of trophic position to analyze the hypothesis of an increase of trophic position with depth as a consequence of the Vinogradov's ladder of migration[16,21,22].

**Respiration.** Specific respiration rates (d$^{-1}$) were derived from the measurement of the ETS activity[54] in those samples from the Malaspina cruise with enough biomass at bathypelagic depths to obtain a subsample and perform the enzymatic assay (Supplementary Table 3). Details of the method can be found elsewhere[54,55]. ETS activity was converted to respiration using a respiration to ETS ratio[55] of 0.5 and using the Arrhenius equation to correct for in situ temperature. A dry weight to protein ratio of 2.49 was used to convert specific respiration rates in a protein to a dry weight basis[56]. Carbon was estimated as 40% of dry weight as in previous studies[51].

**Acoustics.** Backscatter intensity was measured using a LADCP system equipped with two 300 kHz Teledyne/RDI Workhorses, which were mounted on a rosette sampler and deployed at each cast. This allowed to acquire backscatter data below 1000 m depth, thereby extending acoustic assessments of ocean biomass down to the deep sea. Both narrow band units were run in master/slave mode, one upward looking (slave) and one downward (master). LADCP data were processed with IFM-GEOMAR LADCP/LDEO V10 software developed at Columbia University[57]. Data from master/slave transducers were averaged every 10 m to obtain the backscatter profile through the water column (4000 m depth).

**Net primary production.** Surface integrated primary production was downloaded from the Ocean Productivity web site (http://www.science.oregonstate.edu/ocean.productivity/index.php)[58] using the Vertical Generalized Production Model and averaged as in Irigoien et al.[8]. Primary production on an annual basis was obtained from Longhurst et al.[14].

**Statistical analysis.** We used the Moran Index for spatial autocorrelation included in the R statistical software[59] to evaluate the existence of spatial autocorrelation for biomass data along the Malaspina Circumnavigation Expedition. The function lm.morantest() was used[60] and no significant spatial autocorrelations were observed. Statistica v7.0 was also used for general statistical analysis of data.

**Reporting summary**. Further information on research design is available in the Nature Research Reporting Summary linked to this article.

## Data availability

Data sets generated during and/or analyzed during the current study are available at PANGAEA (http://www.pangaea.de). The zooplankton biomass from the Malaspina cruise is available in https://doi.org/10.1594/PANGAEA.922974, the deep-sea zooplankton biomass reviewed is available in https://doi.org/10.1594/PANGAEA.923149, the reviewed carbon flux and respiration are available in https://doi.org/10.1594/PANGAEA.923832, the acoustic backscatter is available in https://doi.org/10.1594/PANGAEA.922619, and the stable isotope data in https://doi.org/10.1594/PANGAEA.919314.

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

## Acknowledgements

This study was funded by projects Malaspina (CSD2008-00077), Bathypelagic (CTM2016-78853-R) from the Spanish Ministry of Economy and Competitiveness, and to projects SUMMER (Grant Agreement 817806) and TRIATLAS (Grant Agreement 817578) from the European Union (Horizon 2020 Research and Innovation Programme). We acknowledge J.C. Garijo for his contribution to the Malaspina zooplankton analysis and A. Santana Del Pino for his advice in statistical procedures. Thanks are also due to all those who contributed to the cruise and sampling during the Malaspina 2010 Expedition, especially the crew of *R.V. "Hespérides"* and the UTM technical staff.

## Author contributions

C.M.D. designed and promoted the Malaspina project. S.H.L. and X.I. conceived this specific project about deep-sea zooplankton. E.F.-N. collected, processed, and ensured data quality of the LADCP. A.B. and C.M. measured zooplankton isotopes. R.K. provided an important collection of deep sea zooplankton data for the review. S.H.L., P.O., A.B., F.E., M.L.F.P., J.I.G.G., A.C., and J.L.A. ensured the quality of the data collection during the Malaspina Circumnavigation, and S.A. provided additional data on gross primary production and acted as cruise leader during the Malaspina project. All authors contributed to interpretation and writing of the paper as well as discussion of results and comments on the manuscript. Figure 4 was produced using elements created by the author.

## Competing interests

The authors declare no competing interest.
