## [Peer Review File · Nature Communications]

Reviewers' Comments:

Reviewer #1:

Remarks to the Author:

This paper convincingly demonstrates that regions of high primary productivity can support large communities of zooplankton, not only in the surface but throughout the water column all the way down to many thousand meters. This is a notion that has been posited before, but this is the first time that I am aware that it has been systematically demonstrated on a global scale. While this is the main result, the bulk of the paper seeks to explore the implications of these findings for our understanding of the biological pump and deep sea ecosystems. This analysis contains many valid points, but is presented in a somewhat clumsy fashion that does a disservice to the author's scholarship.

I cannot recommend publication in its present form, but would encourage the authors to commit to a comprehensive re-write, that would clarify why the manuscript should be of interest to a wide readership.

The manuscript could stand with a reorganization along the lines of the two main questions (at least in my mind) that the observations elicit. Namely

- 1) On a global scale, can we explain how this large biomass of zooplankton is maintained given our current understanding of export flux and respiratory demands placed on it by deep sea bacteria? (apparently not)
- 2) Given that areas of high PP support a disproportionately large biomass of deep sea zooplankton (Fig 2), what are the processes that make the vertical export of organic material so much more efficient in these regions? This is what the title appears to allude to and features in the abstract, but is hardly mentioned in paper.

With regards these, the ms could stand a schematic figure illustrating the pools and processes involved in the carbon budget; net PP, export flux, meso- and bathy-pelagic biomass of prokaryotes & zooplankton and their respiration demands, the three flux components; the slow sinking of POC / marine snow (traditional POC pump), rapid particle flux (fresh biogenic) and the migration ladder. This might help the authors focus on mechanisms that are regionally more or less important in terms of the efficiency of the delivery of organic material to the deep sea. It may also reveal more aspects of the work that have gone unremarked on. For instance, the isotope work that reveals an increase in trophic level with depth is not particularly interesting or surprising. But if you were able to reveal a systematic trend, that under high PP regions this shift in trophic position occurs more rapidly (or slowly for that matter) with depth, that would be an interesting observation that could potentially give greater insight into the processes involved.

A few minor points.

The concept of "carnivorous" copepod is a bit of an antique phrase. Given the vast prevalence of mixotrophy as a trophic mode in protists, any attempted distinction between herbivory, omnivory and carnivory in their predators is pretty much meaningless.

In the supplementary material (Table 1), I gather the fit equation to the vertical distribution is something like

$$\ln B = a z + b$$

where B is the zooplankton biomass abundance, z is the depth in km and a and b are fit parameters. The way this is reported in the table could be very easily misinterpreted.

Reviewer #2:

Remarks to the Author:

This study investigates the relationship between NPP in surface waters of the global ocean and zooplankton biomass in the deeper layers beneath. Using a large amount of data, both new (from the Malaspina Circumnavigation Expedition) and existing (available deep-sea zooplankton biomass), the authors report positive relationships between these two parameters, which they interpret as indicating that increased productivity in the upper layer leads to a greater amount of organic carbon being actively transferred to the deep ocean by zooplankton, augmenting the flux of passively sinking material.

In general, this is an interesting study based on a large amount of data. However, the potential significance of the work and the extent to which it is/is not consistent with other existing estimates of carbon export/CO₂ drawdown (and why this may be in the case of inconsistency) is not clear. Moreover, there seem to be some inconsistencies in the NPP vs. zooplankton biomass relationships that need to be clarified. I have elaborated on my concerns below.

First, it is not clear to me whether the finding that increased productivity in the upper layer leads to a greater amount of organic carbon being actively transferred to the deep ocean by zooplankton implies that previous estimates of carbon export based on flux measurements should be taken as underestimates. In other words, does the zooplankton-driven carbon flux hypothesized here constitute a quantity of carbon export that is currently not accounted for? If this is indeed what the authors are implying, how should it be understood in the context of independent geochemical and/or model-based estimates of carbon export and CO₂ drawdown (i.e., NCP)? If one of the implications of this work is that the ocean's biological pump is potentially stronger than we currently estimate, this needs to be better articulated and more clearly justified/supported in the context of different forms of data/other existing estimates based on other methods (e.g., how do the respiration rates estimated here compare to independent carbon sequestration estimates?). If, on the other hand, I am misunderstanding and this is not an implication of the work, that too needs to be better articulated. In this case, the significance of the findings is less clear to me.

Second, while zooplankton biomass appears to be positively correlated with NPP in Fig. 2, this is not the case for the upper layers in Fig. S1. I cannot work out whether the Malaspina data are included in Fig. 2 or not. In the case that they are not, the fact that the zooplankton data (here presented as backscatter) do not show a correlation with NPP is problematic as it seems to undermine the authors' conclusion that the relationship is robust and almost ubiquitous across the global ocean. I find the idea of Vinogradov's ladder of migration to be compelling, but currently I am unconvinced that it is supported by the data.

Third, how important could ocean circulation be for this analysis? Circulation is not mentioned at all, with the authors assuming processes occurring in the surface are coupled to those occurring in the underlying waters. This assumption needs to be justified. For instance, are the biomes over the authors average large enough to smooth out the potential complications of circulation, bearing in mind that waters in the different depth layers will be transported at different rates and along different paths.

Minor comments

The authors need to define the epi-, meso-, and bathypelagic layers early in the manuscript. This is currently done partway through after these terms have already been used and leads to confusion.

L59-68: I find the opening paragraph repetitive; sentence two and three could be combined.

L71, 72-73 and elsewhere: I encourage the authors to steer clear of the word "believe" as it is not particularly scientific, as well as to limit their use of "hitherto".

L93: what are the units?

L98: replace "at the highest NPP" with "coincident with the highest NPP"; the measurements are correlated, you cannot know at this stage whether NPP is driving the zooplankton biomass differences.

L122-126: How does this relate to/what are the implications for the N isotope data? In general I find the isotope data to be compelling and supportive of the arguments made in the manuscript.

L128-129: What are the implications for the N isotope data?

The paragraph ending on L143 seems to me to need a concluding sentences – why is the content of the paragraph meaningful/important?

L145: How much larger?

L146: Are these decimal places statistically significant?

L216: It seems like a factor of 4 rather than an order of magnitude.

I find the last paragraph to be repetitive while not really clarifying the significance of the findings.

Methods L368-369: Why was this assumption made and how valid is it?

L393-395: Where did the high values come from? What was the motivation for leaving them out?

L426: The word "data" is plural.

Figure 2: Some estimate of error is required, either on the plot or with the regression statistics.

Figure 3: For the lower panel, what do the units mean? It would also be helpful if the regions shown in Fig. S1 were indicated on the panel (epi-, meso-, bathypelagic).

Figure S1: How statistically strong are these relationships? Why are the 0-200 m and 200-1000 m layers so different from the deeper layers. This seems to me potentially highly problematic given that most carbon sequestration should occur in this depth range (particularly 200-1000 m); please see my comment above.

Reviewer #3:

Remarks to the Author:

In this study the authors show that zooplankton biomass in upper, middle and deep ocean water layers are positively correlated with primary productivity in the upper water layer at a global scale. Estimation of respiration rates exceed passive carbon transport through sedimentation. Thus, the authors conclude that active transport through zooplankton migration is a major mechanism for carbon transport into deep ocean layers.

I enjoyed reading the paper and found the study well written and interesting because of the extensive dataset presented and coherent patterns in different ocean basins at a global scale. The biomass observations in combination with carbon demand estimation in different vertical strata of the oceans are convincing that an active mechanism is required, confirming previous suggestions. This study is topical and the results are intriguing for understanding the role of ocean functioning for carbon sequestration.

That said, I have few issues the authors should consider and aspects of the work that can be

improved, which are however not major and are not changing the conclusions.

The authors use stable isotope, SI, to infer trophic position of zooplankton and report an increase with trophic position with depth, up to a trophic level of 5 for deep-water zooplankton. The d15N and trophic position estimation should be, however, treated with care given that bulk SI has limitations: i) baseline shifts, which can be expected with changing species or particle composition with depth; ii) different water masses at different depth layers that may have different d15N signatures; also iii) the protist trophic level is invisible in plankton, see (Gutiérrez-Rodríguez et al. 2014). Some of these issues are outlined here (Sherwooda et al. 2011). In any case, the authors could discuss the potential prey type of the carnivorous copepods and the trophic steps in between that puts them at such high trophic levels. I wonder if another possible mechanism could be that zooplankton feed on sedimenting particles that contains degraded material of consumers of higher trophic levels.

L 51: '.... comparable to the magnitude of passive carbon sequestration ...' – this sentence can be improved to better highlighting the discrepancy between carbon demand and carbon availability through passive sinking particles.

L 96 and related method section: have the authored considered and tested for spatial autocorrelation in the regression approach? This will not change the regression and outcome, but may affect the p-value.

L 182: remove one 'the'

L398: please clarify if these very several individuals of the same species or grouped by size class; also were the carnivorous zooplankton mentioned below sorted separately for SI analysis?

L 406: the term 'carnivores' refers to zooplankton, modify '... zooplankton and carnivorous' to '... zooplankton including carnivorous species' or similar

Table 1: Clarify what the last row refers to, by including 'Total', or similar

Fig. 1: I would suggest to include the regression statistics for the fitted line in the figure legend (or maybe I missed it in the text). Also, it might be useful to highlight in the legend that the x-axis is on log scale.

Fig. 2: same here, mention the log-scale

Fig. 3 legend: I would suggest to improv the legend description to better describe the two graphs. For example, clarify the color gradient and abbreviation 'SV'; describe color coding for the DSL layer; what 'Distance' x-axis, refers to; etc.

References

Gutiérrez-Rodríguez, Andrés, Moira Décima, Brian N. Popp, and Michael R. Landry. "Isotopic Invisibility of Protozoan Trophic Steps in Marine Food Webs." *Limnology and Oceanography* 59, no. 5 1590–98. <https://doi.org/10.4319/lo.2014.59.5.1590>.

Sherwooda, O A, M F Lehmannb, C J Schubert, D B Scottd, and M D McCarthye. "Nutrient Regime Shift in the Western North Atlantic Indicated by Compound-Specific $\Delta^{15}\text{N}$ of Deep-Sea Gorgonian Corals." *PNAS*, 2011.

Reviewer #1 (Remarks to the Author):

This paper convincingly demonstrates that regions of high primary productivity can support large communities of zooplankton, not only in the surface but throughout the water column all the way down to many thousand meters. This is a notion that has been posited before, but this is the first time that I am aware that it has been systematically demonstrated on a global scale. While this is the main result, the bulk of the paper seeks to explore the implications of these findings for our understanding of the biological pump and deep sea ecosystems. This analysis contains many valid points, but is presented in a somewhat clumsy fashion that does a disservice to the author's scholarship.

I cannot recommend publication in its present form, but would encourage the authors to commit to a comprehensive re-write, that would clarify why the manuscript should be of interest to a wide readership.

We thank the reviewer for the comments and criticisms.

Action. These comments led to major improvements of the manuscript, through a reorganization and rewrite to improve clarity, adding also references to recent findings related to active transport in the ocean, a summary of fluxes, and two new figures explaining the processes involved.

The manuscript could stand with a reorganization along the lines of the two main questions (at least in my mind) that the observations elicit. Namely

1) On a global scale, can we explain how this large biomass of zooplankton is maintained given our current understanding of export flux and respiratory demands placed on it by deep sea bacteria? (apparently not)

The large stock of zooplankton points at gaps in our knowledge, which are not new, but add to those identified earlier on the basis of the assessment of microbial respiration in the deep sea (see refs. 32, 33, and 34) and the large stock of mesopelagic fish biomass (see ref. 14). Indeed, average values of particle flux do not satisfy deep-sea prokaryote and zooplankton respiratory demands (see new Figure 4), which is then likely supplemented by episodic and intense particle organic carbon (POC) export fluxes, dissolved organic carbon (DOC) flux, as well as we argue here, active animal-driven flux. We referenced in the text the observation of large healthy phytoplankton in the deep-sea (ref. 16), which provided evidence for widespread occurrence of fast sinking episodes, and also argued that active flux should provide an important fraction of the organic carbon fueling the large biomass in the deep sea. Areas of large productivity should promote large episodic pulses of POC flux, supporting zooplankton biomass in the deep waters. However, the higher zooplankton biomass found in the deep sea in areas of higher productivity should also be promoted by active organic carbon flux (see answer to the next point raised by the reviewer).

Action: This argument has been strengthened by the synthesis of estimates, depicted in the new Figure 4, showing the range of values for passive, active, and DOC fluxes

toward the meso- and bathypelagic zones derived from a review of estimates reported in the literature. This synthesis indicates that respiration by prokaryotes and zooplankton is higher than particle fluxes, so active flux by zooplankton and micronekton should explain, at least in part, the imbalance. Also additional text was added in lines 192-195 to explain this imbalance.

2) Given that areas of high PP support a disproportionately large biomass of deep sea zooplankton (Fig 2), what are the processes that make the vertical export of organic material so much more efficient in these regions? This is what the title appears to allude to and features in the abstract, but is hardly mentioned in paper.

Indeed, this is the case, and we now more explicitly address this question by reporting and providing evidence that the active flux is disproportionately larger, compared to the gravitational sinking POC flux, in high productive areas.

The manuscript provides evidence for the high zooplankton biomass in the deep sea in areas of higher primary production, which we now explain based on a synthesis of existing knowledge of the biological pump. We argue that in addition to gravitational POC export, an active flux driven by vertically-migrant animals should also shunt organic matter to the deep sea, providing food for zooplankton and promoting their relatively large biomass.

We revised the Vinogradov's ladder of migration concept (see new Figure 5) as a possible explanation for this observation. It is clear that the mechanisms involved in such ladder are poorly documented, due to observational challenges, and research effort is required to clarify the potential role of zooplankton and micronekton in promoting an active flux through this ladder. Recently, Hernández-León et al. (2019) (ref. 15) found much higher values of active C flux in productive zones. They observed that POC flux was about 75% of total flux (POC + active fluxes) in oligotrophic zones, in agreement with recent models, whereas in productive waters such as the Guinea Dome or the oceanic upwelling off Northwest Africa, POC flux was only about 20% of total flux (see their Figure 10). Most of the flux in those productive areas was due to zooplankton and micronekton. This observation also matched the observation by Stukel et al. (2018) (ref. 27) who found that almost all the flux in the Costa Rica Dome was due to the migrant fauna. Hernández-León et al. (2019) argued that the constant C supply by the long-lived large fauna (zooplankton and micronekton) and their large individual biomass promoted an additional, active pathway for the biological pump. They also found a relationship between total active flux and primary production, and the slope of this relationship was much higher than that between POC flux and primary production. Although the evidence available does not provide a definitive answer to our observation, it supports the notion of a higher carbon transport in the productive zones of the ocean supported by the active flux by a large migratory biomass.

Action: This explanation is offered, along with the supporting evidence summarized above, in the revised version of the text. In particular, we added text explaining this in lines 138-142 as well as 228-232, along with references to existing estimates.

With regards these, the ms could stand a schematic figure illustrating the pools and processes involved in the carbon budget; net PP, export flux, meso- and bathy-pelagic

biomass of prokaryotes & zooplankton and their respiration demands, the three flux components; the slow sinking of POC / marine snow (traditional POC pump), rapid particle flux (fresh biogenic) and the migration ladder. This might help the authors focus on mechanisms that are regionally more or less important in terms of the efficiency of the delivery of organic material to the deep sea. It may also reveal more aspects of the work that have gone unremarked on. For instance, the isotope work that reveals an increase in trophic level with depth is not particularly interesting or surprising. But if you were able to reveal a systematic trend, that under high PP regions this shift in trophic position occurs more rapidly (or slowly for that matter) with depth, that would be an interesting observation that could potentially give greater insight into the processes involved.

We agree this schematic will indeed help clarify the arguments.

Action: We added a schematic figure (Figure 4), as suggested by the reviewer, comparing vertical fluxes and respiratory carbon demands in oligotrophic and productive areas. The fluxes reported are based on a synthesis of data given in the new Supplementary Table 6 derived from a review of the literature on the carbon supply and demands in the meso- and bathypelagic oceans. As shown in Figure 4, prokaryote plus zooplankton carbon demands exceed POC flux in both oligotrophic and productive areas. Active flux by zooplankton and micronekton helps explain much, but not all, of the carbon deficiency (i.e. Respiratory demands – gravitational POC flux) in the mesopelagic zone in both oligotrophic and productive zones. Thus, active C flux, together with episodic events of high POC flux (more frequent in productive zones) can help support the higher zooplankton biomass in productive zones. However, no data on zooplankton and micronekton carbon flux towards the bathypelagic zone is available at present but, similarly to the mesopelagic zone, the migrant fauna is expected to provide a fraction of the carbon supply. If the migrant fauna contributes to fuel the carbon demand in the bathypelagic zone in a similar proportion as it does in the mesopelagic zone, the flux due to these organisms could partly explain the unbalance.

We have now added new evidence in relation to the isotopic evidence. Specifically, in the new Supplementary Fig. 3 and Supplementary Table 4 we now show the differences between the significant enrichment in $\delta^{15}\text{N}$ with depth for the Trades biome, while there was no significant increase in the more productive Coastal biome. These results are in line with those reported for ecosystems with large seasonal variations in productivity (Romero-Romero et al., 2019) (ref. 25) and attributed to changes in the relative importance of recycled nitrogen at the base of the food web through the microbial and microzooplankton pathways (e.g. Basedow et al., 2016) (ref. 26).

A few minor points.

The concept of “carnivorous” copepod is a bit of an antique phrase. Given the vast prevalence of mixotrophy as a trophic mode in protists, any attempted distinction between herbivory, omnivory and carnivory in their predators is pretty much meaningless.

We agree that a prior classification of copepods in trophic types is an oversimplification, particularly in deep layers where in situ herbivory is expected to be

negligible. The purpose of grouping individuals in the same trophic category was to obtain enough samples for stable isotope analysis.

Action: The estimations of trophic position were removed as well as the corresponding text.

In the supplementary material (Table 1), I gather the fit equation to the vertical distribution is something like $\ln B = a z + b$ where B is the zooplankton biomass abundance, z is the depth in km and a and b are fit parameters. The way this is reported in the table could be very easily misinterpreted.

Agreed.

Action: The equation is now stated in the Table legend: $\ln B = a z + b$

Reviewer #2 (Remarks to the Author):

This study investigates the relationship between NPP in surface waters of the global ocean and zooplankton biomass in the deeper layers beneath. Using a large amount data, both new (from the Malaspina Circumnavigation Expedition) and existing (available deep-sea zooplankton biomass), the authors report positive relationships between these two parameters, which they interpret as indicating that increased productivity in the upper layer leads to a greater amount of organic carbon being actively transferred to the deep ocean by zooplankton, augmenting the flux of passively sinking material.

In general, this is an interesting study based on a large amount of data. However, the potential significance of the work and the extent to which it is/is not consistent with other existing estimates of carbon export/CO₂ drawdown (and why this may be in the case of inconsistency) is not clear. Moreover, there seem to be some inconsistencies in the NPP vs. zooplankton biomass relationships that need to be clarified. I have elaborated on my concerns below.

First, it is not clear to me whether the finding that increased productivity in the upper layer leads to a greater amount of organic carbon being actively transferred to the deep ocean by zooplankton implies that previous estimates of carbon export based on flux measurements should be taken as underestimates. In other words, does the zooplankton-driven carbon flux hypothesized here constitute a quantity of carbon export that is currently not accounted for? If this is indeed what the authors are implying, how should it be understood in the context of independent geochemical and/or model-based estimates of carbon export and CO₂ drawdown (i.e., NCP)? If one of the implications of this work is that the ocean's biological pump is potentially stronger than we currently estimate, this needs to be better articulated and more clearly justified/supported in the context of different forms of data/other existing estimates based on other methods (e.g., how do the respiration rates estimated here compare to independent carbon sequestration estimates?). If, on the other hand, I am misunderstanding and this is not an implication of the work, that too needs to be better articulated. In this case, the significance of the findings is less clear to me.

We agree with the reviewer that the implication and findings need to be presented with greater clarity, as also requested by Reviewer #1.

Indeed, the gravitational sinking carbon flux (i.e. estimates based on the measurement of sinking of particles), by far the most studied mechanism of vertical carbon transport, does not suffice to meet the respiratory demands of prokaryotes and zooplankton in the meso- and bathypelagic layers of the ocean, as had been reported already in the past (see references in the text).

In our study, we report that zooplankton biomass in the meso- and bathypelagic layers increases with primary production in the euphotic zone. This observation suggests a higher vertical carbon export, through gravitational and active fluxes, in those productive areas. Although the former is large in productive areas, the latter should also be potentially higher given the higher zooplankton biomass in the deep sea. This is partially explained by the higher active flux reported in highly productive areas such as the Guinea Dome and the oceanic upwelling off Northwest Africa. Recently, Hernández-León et al. (2019) (ref. 15) found much higher values of active C flux in productive zones. They observed that POC flux was about 75% of total flux (POC + active fluxes) in oligotrophic zones, in agreement with recent models, whereas in productive waters such as the Guinea Dome or the oceanic upwelling off Northwest Africa, POC flux was only about 20% of total flux (see their Figure 10). Most of the flux in those productive areas was due to zooplankton and micronekton. This observation also matched the observation by Stukel et al. (2018) (ref. 27) who found that almost all the flux in the Costa Rica Dome was due to the migrant fauna. Hernández-León et al. (2019) argued that the constant C supply by the long-lived large fauna (zooplankton and micronekton) and their large individual biomass promoted an additional, active pathway for the biological pump. They also found a relationship between total active flux and primary production, and the slope of this relationship was much higher than that between POC flux and primary production. Although the evidence available does not provide a definitive answer to our observation, it supports the notion of a higher carbon transport in the productive zones of the ocean supported by the active flux by a large migratory biomass.

Action: We have revised the text to improve clarity, conducted a synthesis of estimates of carbon fluxes and demands (Supplementary Table 6), and added a new Figure 4 where we illustrate the processes involved and their magnitude. These estimates show that respiratory demands exceed the sum of the POC flux, DOC flux, and active flux. Thus, the results indicate that not only the active flux is an important component of the vertical carbon flux, but that very likely the gravitational sinking flux is underestimated, possibly due to underrepresenting episodic sinking events. This explanation is offered, along with the supporting evidence summarized above, in the revised version of the text. In particular, we added a text explaining this in lines 138-142 as well as 228-232.

Second, while zooplankton biomass appears to be positively correlated with NPP in Fig. 2, this is not the case for the upper layers in Fig. S1. I cannot work out whether the Malaspina data are included in Fig. 2 or not. In the case that they are not, the fact that the zooplankton data (here presented as backscatter) do not show a correlation with NPP is problematic as it seems to undermine the authors' conclusion that the relationship is robust and almost ubiquitous across the global ocean. I find the idea of Vinogradov's ladder of migration to be compelling, but currently I am unconvinced that it is supported by the data.

Figure 2 is only built using the zooplankton biomass from the Malaspina Expedition (this is now stated in the Figure legend). This data was obtained using a Multinet sampler at different layers from the epi- to the bathypelagic zones. Supplementary Figure 1 was also built using the backscatter strength of the Lowered Acoustic Doppler Current Profiler (LADCP) attached to the rosette sampler during the Malaspina cruise. As also observed in the biomass data, a large variability was found in the epipelagic layer (0-200 m depth) in relation to primary production because of the mismatch between measuring biomass or backscatter at a given time during the cruise and the remote sensing average used for primary production in a rather highly dynamic system as the epipelagic zone. The highest primary production was observed in the coastal domain (mainly close to upwelling zones) but these high values were not quite coupled to high backscatter in the epipelagic zone or even in the mesopelagic zone. The strength of correlations increase with depth, possibly because of a slower turnover of zooplankton biomass at depth that may provide less fluctuations and a better match with average primary production. Hence, the weaker correlation between instantaneous zooplankton biomass and average primary production for the epipelagic zone is not unexpected and does not weaken the argument.

Action: We address this in the new version of the manuscript (lines 228-241, see Suppl. Fig. 1 legend). These relationships show a strong relationship in deep waters (meso- and bathypelagic zones), characterized by a higher stability with biomass and backscatter average matching primary production. In contrast, the link between average primary production and instantaneous (at the time of sampling) zooplankton biomass in the epipelagic zone is expected to be more variable, and require assessment through dynamic time series, incorporated lag times, or through averages in both primary production and zooplankton biomass. However, the zooplankton biomass in the mesopelagic and bathypelagic is likely to be less variable over time and better reflect the average primary production in the epipelagic zone. To our view, we do not think this undermines our conclusions of higher biomass in the deep sea mirroring primary production. We also clarified the sources of data from the figure legend (Fig. 2). We added text clarifying this issue in lines 228-241.

Third, how important could ocean circulation be for this analysis? Circulation is not mentioned at all, with the authors assuming processes occurring in the surface are coupled to those occurring in the underlying waters. This assumption needs to be justified. For instance, are the biomes over the authors average large enough to smooth out the potential complications of circulation, bearing in mind that waters in the different depth layers will be transported at different rates and along different paths.

The reviewer is correct in that ocean circulation maybe important and it could also help explain the lower (biomass) or lack (backscatter) of correlation in the epipelagic zone (see Suppl. Fig. 1 legend), particularly so for upwelling domains, that are highly dynamic. The epipelagic zone is much more dynamic than the deep sea and, therefore, it is expected a much higher variability comparing primary production and biomass (or backscatter). In particular, as current velocity and transport in the epipelagic could be an order of magnitude faster than in the ocean interior, productive areas maybe exporting zooplankton biomass horizontally towards the open ocean, further contributing to a

weak relationship between average primary production and primary production observed in the epipelagic zone.

Action: We have now added text discussing this aspect in lines 228-241.

Minor comments

The authors need to define the epi-, meso-, and bathypelagic layers early in the manuscript. This is currently done partway through after these terms have already been used and leads to confusion.

Action: We agree and now define these layers the first time they are named in the manuscript.

L59-68: I find the opening paragraph repetitive; sentence two and three could be combined.

Action: Agreed. Done.

L71, 72-73 and elsewhere: I encourage the authors to steer clear of the word “believe” as it is not particularly scientific, as well as to limit their use of “hitherto”.

Action: Agreed. Done.

L93: what are the units?

Units are km^{-1} .

Action: This is stated there (now line 96) and in the Suppl. Table 1.

L98: replace “at the highest NPP” with “coincident with the highest NPP”; the measurements are correlated, you cannot know at this stage whether NPP is driving the zooplankton biomass differences.

Action: Agreed. Done.

L122-126: How does this relate to/what are the implications for the N isotope data? In general I find the isotope data to be compelling and supportive of the arguments made in the manuscript.

Action: We modified this paragraph by adding the results in a new Supplementary Fig. 3 and Supplementary Table 4, and added a clarification in the text (now line 136): “In addition to *in situ* feeders, bathypelagic zooplankton communities include seasonal ontogenetic migrants overwintering at depth and not feeding there (ref. 7), and organisms using the deep-sea as a refuge from predators (ref. 21).”

L128-129: What are the implications for the N isotope data?

We agree this need to be elaborated further.

Action: We agree and have now added new evidence in relation to the isotopic evidence. Specifically, in the new Supplementary Fig. 3 and Supplementary Table 4 we now show the differences between the significant enrichment in $\delta^{15}\text{N}$ with depth for the Trades biome, while there was no significant increase in the more productive Coastal biome. These results are in line with those reported for ecosystems with large seasonal variations in productivity (Romero-Romero et al., 2019) (ref. 25) and attributed to changes in the relative importance of recycled nitrogen at the base of the food web through the microbial and microzooplankton pathways (e.g. Basedow et al., 2016) (ref. 26). In productive areas the potentially higher episodic sedimentation (e.g., Agustí et al., 2015) (ref. 16) and the stronger active flux performed by zooplankton and micronekton (e.g., Stukel et al., 2018 (ref. 27); Hernández-León et al., 2019 (ref. 15)) should promote, jointly with vertical migration, the homogenization of stable isotopes in the water column.

The paragraph ending on L143 seems to me to need a concluding sentences – why is the content of the paragraph meaningful/important?

Agreed.

Action: We combined this and the previous paragraph in order to better explain the findings derived from stable isotopes data in the context of the downward flux.

L145: How much larger?

As stated in the Introduction of the manuscript, the zooplankton biomass in the deep sea was assumed to be negligible (ref. 13). We show here that this biomass change over one order of magnitude in the deep sea in areas of increased productivity (see Figure 2).

L146: Are these decimal places statistically significant?

Agreed.

Action: We rounded the figures for clarity.

L216: It seems like a factor of 4 rather than an order of magnitude.

Figure 2 shows that meso- and bathypelagic biomass increases an order of magnitude from low to high primary production.

I find the last paragraph to be repetitive while not really clarifying the significance of the findings.

Action: We modified the paragraph now as a concluding remark.

Methods L368-369: Why was this assumption made and how valid is it?

Most of the data provided by the German researchers split zooplankton biomass in larger and smaller than 5 mm. Thus, we adopted the same criteria for most of data when this information was available.

Action: This is now reported in methods. The text reads: “We reviewed 274 profiles (Supplementary Fig. 4) of zooplankton biomass smaller than 5 mm when this information was available...”

L393-395: Where did the high values come from? What was the motivation for leaving them out?

We decided now to maintain these data related to the Pacific oxygen minimum zone in the regression and the correlation improved as previously observed. We also analyze for outliers and they were not considered.

L426: The word “data” is plural.

Action: Corrected.

Figure 2: Some estimate of error is required, either on the plot or with the regression statistics.

Action: The regression statistics are now given in the text, in lines 99-101.

Figure 3: For the lower panel, what do the units mean? It would also be helpful if the regions shown in Fig. S1 were indicated on the panel (epi-, meso-, bathypelagic).

Values are backscatter volume strength (SV) in decibels.

Action: The units, although given in the scale, are now given in the Figure legend for clarity. Other information as the research vessel transect and the vertical dashed lines are now explained.

Figure S1: How statistically strong are these relationships? Why are the 0-200 m and 200-1000 m layers so different from the deeper layers. This seems to me potentially highly problematic given that most carbon sequestration should occur in this depth range (particularly 200-1000 m); please see my comment above.

As indicated above, we now explain that the temporal variability in zooplankton biomass is likely to be much greater in the epipelagic zone, so that the relationship between mean NPP and instantaneous epipelagic zooplankton biomass (at the time of sampling) is expected to be noisier than in the more stable mesopelagic and bathypelagic layers, where zooplankton biomass maybe in near steady state and therefore show a stronger relationship to average NPP. In addition, horizontal transport and export could be one order of magnitude larger in the epipelagic layer, further adding noise to the relationship between average primary production and zooplankton biomass in the epipelagic.

Action: The regression statistics are given in the text, now in lines 107-109. The question of the differences between the epi- and mesopelagic backscatter in relation to

the bathypelagic was answered above. In any case, it is now explained in the figure legend.

Reviewer #3 (Remarks to the Author):

In this study the authors show that zooplankton biomass in upper, middle and deep ocean water layers are positively correlated with primary productivity in the upper water layer at a global scale. Estimation of respiration rates exceed passive carbon transport through sedimentation. Thus, the authors conclude that active transport through zooplankton migration is a major mechanism for carbon transport into deep ocean layers.

I enjoyed reading the paper and found the study well written and interesting because of the extensive dataset presented and coherent patterns in different ocean basins at a global scale. The biomass observations in combination with carbon demand estimation in different vertical strata of the oceans are convincing that an active mechanism is required, confirming previous suggestions. This study is topical and the results are intriguing for understanding the role of ocean functioning for carbon sequestration.

Agreed. We thank the reviewer for her/his comments.

That said, I have few issues the authors should consider and aspects of the work that can be improved, which are however not major and are not changing the conclusions.

The authors use stable isotope, SI, to infer trophic position of zooplankton and report an increase with trophic position with depth, up to a trophic level of 5 for deep-water zooplankton. The $\delta^{15}\text{N}$ and trophic position estimation should be, however, treated with care given that bulk SI has limitations: i) baseline shifts, which can be expected with changing species or particle composition with depth; ii) different water masses at different depth layers that may have different $\delta^{15}\text{N}$ signatures; also iii) the protist trophic level is invisible in plankton, see (Gutiérrez-Rodríguez et al. 2014). Some of these issues are outlined here (Sherwooda et al. 2011). In any case, the authors could discuss the potential prey type of the carnivorous copepods and the trophic steps in between that puts them at such high trophic levels. I wonder if another possible mechanism could be that zooplankton feed on sedimenting particles that contains degraded material of consumers of higher trophic levels.

We acknowledge the limitations of the SI-bulk analysis. However, the low amount of sample available in this case did not allow more sophisticated determinations (e.g. compound specific isotope analysis) that would have provided better estimations of potential trophic position (TP) change with depth.

Action: In the revised version we removed the estimations of TP and illustrated the link between deep enrichment and surface productivity using only bulk $\delta^{15}\text{N}$ values. In the new Supplementary Fig. 3 and Supplementary Table 4 we now show the differences between the significant enrichment in $\delta^{15}\text{N}$ with depth for the Trades biome while there was no significant increase in the more productive Coastal biome. These results are in line with those reported for ecosystems with large seasonal variations in productivity (Romero-Romero et al., 2019) (ref. 25) and attributed to changes in the relative

importance of recycled nitrogen at the base of the food web through the microbial and microzooplankton pathways (e.g. Basedow et al., 2016) (ref. 26).

L 51: ‘ comparable to the magnitude of passive carbon sequestration ...’ – this sentence can be improved to better highlighting the discrepancy between carbon demand and carbon availability through passive sinking particles.

We agree.

Action: We now clarify better this issue in the new Figure 4, including a Figure to illustrate the problem. We reviewed POC flux, active flux, prokaryote and zooplankton respiration in order to illustrate the unbalance and whether the unknown active sequestration flux could, at least in part, fill this unbalance.

L 96 and related method section: have the authored considered and tested for spatial autocorrelation in the regression approach? This will not change the regression and outcome, but may affect the p-value.

We agree this need be verified. We now used the Moran Index included in the R statistical software to evaluate the existence of spatial autocorrelation for biomass data along the Malaspina Circumnavigation Expedition. The function `lm.morantest()` was used and no significant spatial autocorrelations were observed.

Action: This is now reported in the methods section.

L 182: remove one ‘the’

Done.

L398: please clarify if these very several individuals of the same species or grouped by size class; also were the carnivorous zooplankton mentioned below sorted separately for SI analysis?

Action: The text was modified: “Several individuals were combined to obtain at least 10 μg N per sample. In most cases these combined samples included several species or genera of the same trophic category according to the literature (Benedetti et al. 2016) (ref. 49).

L 406: the term ‘carnivores’ refers to zooplankton, modify ‘... zooplankton and carnivorous’ to ‘ ... zooplankton including carnivorous species’ or similar

Action: The estimations of TP were removed as well as the corresponding text.

Table 1: Clarify what the last row refers to, by including ‘Total’, or similar

Action: Agreed. Done.

Fig. 1: I would suggest to include the regression statistics for the fitted line in the figure legend (or maybe I missed it in the text). Also, it might be useful to highlight in the legend that the x-axis is on log scale.

Action: Done.

Fig. 2: same here, mention the log-scale

Action: Done.

Fig. 3 legend: I would suggest to improve the legend description to better describe the two graphs. For example, clarify the color gradient and abbreviation ‘SV’; describe color coding for the DSL layer; what ‘Distance’ x-axis, refers to; etc.

Action: Done.

References

Gutiérrez-Rodríguez, Andrés, Moira Décima, Brian N. Popp, and Michael R. Landry. “Isotopic Invisibility of Protozoan Trophic Steps in Marine Food Webs.” *Limnology and Oceanography* 59, no. 5 1590–98. <https://doi.org/10.4319/lo.2014.59.5.1590>.

Sherwooda, O A, M F Lehmannb, C J Schubert, D B Scottd, and M D McCarthye. “Nutrient Regime Shift in the Western North Atlantic Indicated by Compound-Specific $\Delta^{15}\text{N}$ of Deep-Sea Gorgonian Corals.” *PNAS*, 2011.

Reviewers' Comments:

Reviewer #1:

None

Reviewer #2:

Remarks to the Author:

Please see uploaded pdf file (also pasted below).

Recommendation: major revisions required.

Review of "Large deep-sea zooplankton biomass mirrors primary production in the global ocean"

This study uses a vast amount of data on zooplankton biomass at various depths in the deep ocean to show – for the first time – that deep-sea zooplankton biomass is correlated with overlying rates of primary production (on large time- and space scales). Furthermore, the authors calculate that the zooplankton carbon demand, when combined with that estimated for bacteria, is much higher than can be met by the passive flux of particulate organic carbon (POC). This work is definitely important and worthy of publication in a high-impact journal. However, while the manuscript has improved since the last time I reviewed it, I still have concerns (some major) that I feel need to be addressed before I can recommend its publication.

1. Dissolved organic carbon (DOC): A central finding of this study is that the deep-sea carbon demand of bacteria and zooplankton (given the new higher estimates of zooplankton biomass) is ~100% higher than the carbon supplied by gravitational/passive sinking. This means there has to be an additional mechanism (or mechanisms) of carbon supply, which the authors conclude is active transport by zooplankton. However, there is no discussion at all in the main text of the potential role of DOC, which can represent a significant quantity of carbon to bacteria. Strangely, DOC is included in fig. 4 (the new schematic summarizing the carbon fluxes into the deep sea) and is very briefly mentioned in the fig caption, but that is it. The DOC flux has been estimated to be as high as the traditional (i.e., gravitational) POC flux in some regions (which the authors acknowledge in the figure caption, referring the reader to Table S6, although I couldn't find any DOC data there), which would go a long way towards closing the deep-sea carbon budget. This cannot be ignored. To my mind, failing to account for and discuss the influence of DOC completely undermines the authors' conclusion that a "new" (i.e., zooplankton-facilitated) flux of organic matter is required. I need to be convinced that this cannot be supplied by DOC.

Related to the discussion of the putative active flux of carbon, the authors make mention of episodic events of high carbon flux but don't elaborate on what they mean by this or how important they think these events could be. Are they currently not accounted for in estimates of the gravitational POC flux?

2. Implications of the higher deep-sea carbon demand: I raised this point in my first review, and I don't feel that it has been adequately addressed in the most recent version of the manuscript. If a greater amount of organic carbon is being actively transferred into the deep ocean by zooplankton than previously estimated, is the implication that we have underestimated the strength of the biological pump or is it that the carbon being transferred to the deep ocean via the biological pump needs to be differently apportioned (i.e., if more of it is being actively transferred by zooplankton, does that require less be transported by some other mechanism?). This concern can be summed up as: what are the implications for the biological carbon pump and atmospheric CO₂ sequestration in the deep sea?

3. The nitrogen (N) isotope data: I think this may derive from some disagreement between the reviewers as to the importance of the N isotope data, but at present, these are mentioned somewhat randomly in the text (e.g., line 124: "evidence for 15N enrichment", with no definition

of what this means). I recommend a more systematic treatment of these data if they are to be included. More importantly, the data are not shown anywhere, not even in the Supplemental Information. It is my feeling that if the authors plan to rely on these data, they need to be presented somewhere. Since the $\delta^{15}\text{N}$ data are used as a line of evidence for the idea of Vinogradov's ladder of migration, perhaps they could be included in fig. 5. In its current form, I find fig. 5 difficult (I'm actually not convinced that it's necessary) – for instance, the red and blue circles are not explained, and it might be beneficial to include the names of example species on each "rung" of the ladder. Additionally, I recommend removing the black background.

Minor comments

I still feel that the various layers (epi-, meso-, and bathypelagic) need to be defined sooner than they are. I get lost around lines 81-86 without this information.

Line 39-42: I find this sentence difficult – perhaps it has too many clauses/commas. I recommend revisiting it for clarity.

Line 42: what is meant by "independent"?

Line 43: refers to the epi-, meso-, and bathypelagic layers, but later on (e.g., line 55), it's only the bathypelagic?

Line 49: what does "surface layers" mean here?

Line 50: sentence beginning "estimated remineralization..." – this sentence could be simplified for clarity. The authors mean that the carbon demand implied by their zooplankton respiration rates is higher than can be met by existing estimates of the carbon supply to the deep ocean; I find their use of "estimated remineralization" makes the statement vague.

Line 56: replace "and" with "as well as" to improve clarity. Also, "an active transport mechanism associated with vertical..."

Line 69: replace "in terms of" with "relative to".

Line 73: as in my earlier review, I recommend avoiding the use of "believe".

Line 74-76: this sentence needs to be better connected to the one that precedes it. Also, here and elsewhere, the frequent use of "as recently observed" can be removed – the appropriate citation implies this.

Line 80: reference 7 – is the reference for what has been assumed in the past? Or is the reference for the fact that more carbon is being exported? If the former, a reference is required for the statement that more carbon is being exported. If the latter, this study (i.e., ref 7) is nearly 20 years old, which implies that it has long been recognized that more carbon is required. If this is the case, it should be stated/framed thus.

Line 81: the use of "suggest" is inappropriate. Better would be "one implication of these various lines of evidence is..."

Line 83 and elsewhere: I recommend restraint in the use of "hitherto"

Line 84: what does "deep-sea" mean here?

Line 98 onwards: I found some of this paragraph to be repetitive. Also, the insertion of the regression relationships into the text with no definition of LnPP, LnB etc does not provide the

reader with any useful information. Either the relationships need to be better contextualized/explained/defined, or the regression statistics need to be presented differently.

Line 124 onwards: see my comment above about the N isotope data.

Line 142: what observation?

Line 144: the authors have a myriad of ways of referring to the deep ocean, some more specific than others. Here they now use "twilight zone". It's confusing to the reader to have so many different terms used to refer to the same thing with varying degrees of specificity. I recommend being more selective, as also as precise as possible each time – when you say deep sea, for instance, do you mean the entire deep ocean, certain layers, etc.

Line 147: from what? How?

Line 149: there are aspects of this paragraph that are repetitive. I suggest revisiting for clarity and repetition. Likewise, line 166-168 and 173-175.

Line 195: that can be supported by the passive flux? Also, please see my comment above about DOC. Finally, "an important contribution from active carbon flux" is vague – be specific in what you mean by "important".

Line 206: I don't understand this sentence – "fragmentation" (this is the first use of this term), "be the rule" (what is meant by this?)

Line 223: if there have been 50 years of research efforts, a few citations are required here.

Line 229: surface chlorophyll?

Line 234: horizontal transport of what?

Line 245-248: please clarify what you mean here.

Line 258: please see my comment above about the implications of all this for the biological pump.

Methods

Line 418: "published conversion factors" – how much error might this introduce.

Line 455: protein to dry weight conversion – should there be a reference for this? If not, where did the number come from?

I think a little more detail needs to be provided on the Red Sea data that were excluded.

Data availability: I am not aware of the requirements of the journal, but surely the data should be made available in a public repository – the days of "contact the first author" have passed and it's important for the integrity of the work (as per the FAIR data principles), that the data be publicly available. I encourage the authors to take this seriously.

Reviewer #3:

Remarks to the Author:

The authors have done a good job with revising the ms and the additional figures help to better understand the processes. I have only a few additional comments.

L 87: I would suggest that this paragraph includes a short description of the approach taken, like N-Isotopes and respiration, particularly since the methods are given at the end and a big part of the results are moved to the supplementary information. This will help the reader to better follow the results and conclusions.

L 150: change to 'than previously estimated'. Can you include a reference supporting this statement?

L 233: should this be '..., such as the pelagic zone'?

Figure 1: In this figure 'oceanic biome' is the main grouping variable but the biomes have not been described/defined before. A supplementary map showing the regions could be a solution to this.

Figure 2: The figure legend states, 'Note that both axes are in logarithmic scale.' But the x-axis is log with base 2, while the y-axis is log with base 10, even though annotated as if it was log with base 2. Perhaps this can be clarified in the figure legend to avoid confusion or misinterpretation.

Figure 3: Here, alignment of the map with the graph would give a clearer interpretation, and maybe including the oceanic biomes.

Figure 4: Explain what the difference between green and red arrows is to avoid misinterpretation

Figure 5: Clarify what the difference between red and blue migration patterns is.

Supplementary Table 1: Since focus in this table is put on Biome types, it would be interesting to see the statistical significance of this term.

REVIEWER COMMENTS

Reviewer #2 (Remarks to the Author):

Recommendation: major revisions required.

Review of “Large deep-sea zooplankton biomass mirrors primary production in the global ocean”

This study uses a vast amount of data on zooplankton biomass at various depths in the deep ocean to show – for the first time – that deep-sea zooplankton biomass is correlated with overlying rates of primary production (on large time- and space scales). Furthermore, the authors calculate that the zooplankton carbon demand, when combined with that estimated for bacteria, is much higher than can be met by the passive flux of particulate organic carbon (POC). This work is definitely important and worthy of publication in a high-impact journal. However, while the manuscript has improved since the last time I reviewed it, I still have concerns (some major) that I feel need to be addressed before I can recommend its publication.

1. Dissolved organic carbon (DOC): A central finding of this study is that the deep-sea carbon demand of bacteria and zooplankton (given the new higher estimates of zooplankton biomass) is ~100% higher than the carbon supplied by gravitational/passive sinking. This means there has to be an additional mechanism (or mechanisms) of carbon supply, which the authors conclude is active transport by zooplankton. However, there is no discussion at all in the main text of the potential role of DOC, which can represent a significant quantity of carbon to bacteria. Strangely, DOC is included in fig. 4 (the new schematic summarizing the carbon fluxes into the deep sea) and is very briefly mentioned in the fig caption, but that is it. The DOC flux has been estimated to be as high as the traditional (i.e., gravitational) POC flux in some regions (which the authors acknowledge in the figure caption, referring the reader to Table S6, although I couldn't find any DOC data there), which would go a long way towards closing the deep-sea carbon budget. This cannot be ignored. To my mind, failing to account for and discuss the influence of DOC completely undermines the authors' conclusion that a “new” (i.e., zooplankton-facilitated) flux of organic matter is required. I need to be convinced that this cannot be supplied by DOC.

Related to the discussion of the putative active flux of carbon, the authors make mention of episodic events of high carbon flux but don't elaborate on what they mean by this or how important they think these events could be. Are they currently not accounted for in estimates of the gravitational POC flux?

Comment: We agree that DOC need to be addressed, and indeed it will be consistent with the conceptual diagram included in Fig. 4.

Action: We now address the potential role of DOC. The text now reads in lines 186-193:

Carbon export from the photic layer includes a passive and active flux as well as an export of DOC (Fig. 4), estimated to account for ~20% of global passive export production but about half of the total passive organic carbon export in the oligotrophic

subtropical oceans (ref. 35, Roshan and DeVries 2017). However, DOC export does not appear to penetrate beyond the mesopelagic layer, where it supports an estimated DOC, on average, only 8.4% of respiratory demands (ref. 36, Aristegui et al. 2002). Hence, neither passive POC nor DOC fluxes can account for the high carbon demand by mesopelagic and bathypelagic zooplankton and prokaryotes.

Action: DOC flux in Figure 4 is estimated from POC flux. This is changed now in the legend of Figure 4.

2. Implications of the higher deep-sea carbon demand: I raised this point in my first review, and I don't feel that it has been adequately addressed in the most recent version of the manuscript. If a greater amount of organic carbon is being actively transferred into the deep ocean by zooplankton than previously estimated, is the implication that we have underestimated the strength of the biological pump or is it that the carbon being transferred to the deep ocean via the biological pump needs to be differently apportioned (i.e., if more of it is being actively transferred by zooplankton, does that require less be transported by some other mechanism?). This concern can be summed up as: what are the implications for the biological carbon pump and atmospheric CO₂ sequestration in the deep sea?

Comment: We agree that this is an important aspect of the paper that cannot be left ignored, and apologize for having failed to address it fully in the previous revision.

Action: We now address this important aspect. The text now reads in lines 262-272:

Current estimates of the biological pump transferring carbon to the dark ocean embedded in depictions of the global carbon cycle (ref. 48, Ciais et al. 2013) consider passive fluxes alone (i.e. POC and DOC fluxes in Fig. 4). Our estimates that the active organic carbon export from the photic layer must be of a similar magnitude as passive POC +DOC flux to satisfy calculated carbon demands implies that the biological pump exports twice as much carbon to the twilight and dark ocean as included in current depictions of the global carbon budget (ref. 48, Ciais et al. 2013). Moreover, whereas most of the DOC flux is remineralized within the mesopelagic layer (ref. 36, Aristegui et al. 2002), the Vinogradov ladder connects predator-prey active transport chains acting in tandem to delivered a minimum of 0.44 Pg C y⁻¹ into the bathypelagic ocean, where carbon is sequestered over time scales relevant to climate projections.

Hence, we conclude that the large magnitude of deep-sea zooplankton biomass and the conservative estimate of the associated carbon transport into the bathypelagic ocean provided here point at a need to reconsider the oceanic carbon budget to account for the role of pelagic fauna as core components of the biological pump helping to reconcile estimates of carbon supply with those of demand derived from respiratory organic carbon demands by different components of the deep-sea ecosystems^{32,33,34}.

3. The nitrogen (N) isotope data: I think this may derive from some disagreement between the reviewers as to the importance of the N isotope data, but at present, these are mentioned somewhat randomly in the text (e.g., line 124: "evidence for 15N enrichment", with no definition of what this means). I recommend a more systematic treatment of these data if they are to be included. More importantly, the data are not shown anywhere, not even in the Supplemental Information. It is my feeling that if the

authors plan to rely on these data, they need to be presented somewhere. Since the d15N data are used as a line of evidence for the idea of Vinogradov's ladder of migration, perhaps they could be included in fig. 5. In its current form, I find fig. 5 difficult (I'm actually not convinced that it's necessary) – for instance, the red and blue circles are not explained, and it might be beneficial to include the names of example species on each “rung” of the ladder. Additionally, I recommend removing the black background.

Clarifications were added in the main text (lines 93-95 and L 126-127), and in the Methods section (lines 475-477). Also, a new version of Supplementary Figure 3 is now provided.

Moreover, we agree with the reviewer about the need to show Figure 5, so we decided to remove it. For d15N data, Supplementary Figure 3 now provides a reasonable picture of the increase of trophic position with depth.

Minor comments

I still feel that the various layers (epi-, meso-, and bathypelagic) need to be defined sooner than they are. I get lost around lines 81-86 without this information.

We define now these layers the first time they are named. The mesopelagic zone in line 38 “...meso- (200-1000 m layer) and...” The bathypelagic zone in line 38-39 “(1000-4000 m)”, and the epipelagic zone in line 42 “epi- (0-200 m)”.

Line 39-42: I find this sentence difficult – perhaps it has too many clauses/commas. I recommend revisiting it for clarity.

We now deleted the word gravitational as passive flux is widely accepted to define this flux, avoiding in this way some commas. We think this phrase is now much clear.

Line 42: what is meant by “independent”?

Independent refers to, in our case, three different relationships between zooplankton biomass or proxies (biomass in the Malaspina cruise, LADCP acoustics, and reviewed biomass) and primary production. The three relationships are independent as the results of one of them does not affect the others. This is a common definition in statistics.

Line 43: refers to the epi-, meso-, and bathypelagic layers, but later on (e.g., line 55), it's only the bathypelagic?

In line 54 we refer to bathypelagic layer because we estimated remineralization by zooplankton in this layer (lines 50-51), thus we only refer to this layer here and all along the manuscript. For more clarity we added in line 50 the word bathypelagic after deep-sea.

Line 49: what does “surface layers” mean here?

It refers to the euphotic layer where primary production occurs. In order to clarify we deleted “surface” and added the words “euphotic” and “primary” in line 48.

Line 50: sentence beginning “estimated remineralization...” – this sentence could be simplified for clarity. The authors mean that the carbon demand implied by their zooplankton respiration rates is higher than can be met by existing estimates of the carbon supply to the deep ocean; I find their use of “estimated remineralization” makes the statement vague.

The reviewer is right but our remineralization values are estimations as they are not measured because of the obvious impossibility. They were obtained from respiration obtained through ETS activity.

Line 56: replace “and” with “as well as” to improve clarity. Also, “an active transport mechanism associated with vertical...”

Agreed. Done.

Line 69: replace “in terms of” with “relative to”.

Agreed. Done.

Line 73: as in my earlier review, I recommend avoiding the use of “believe”.

Agreed. We change the word “believed” by “assumed”.

Line 74-76: this sentence needs to be better connected to the one that precedes it. Also, here and elsewhere, the frequent use of “as recently observed” can be removed – the appropriate citation implies this.

Agreed. We modified the phrase deleting “as recently observed”.

Line 80: reference 7 – is the reference for what has been assumed in the past? Or is the reference for the fact that more carbon is being exported? If the former, a reference is required for the statement that more carbon is being exported. If the latter, this study (i.e., ref 7) is nearly 20 years old, which implies that it has long been recognized that more carbon is required. If this is the case, it should be stated/framed thus.

The reviewer is right. As it was written it seemed a reference was necessary. However, the proper sense was the second interpretation by the reviewer. Thus, we deleted the words “in the past” and now is written as “... than often assumed” in line 79.

Line 81: the use of “suggest” is inappropriate. Better would be “one implication of these various lines of evidence is...”

Agreed. Done.

Line 83 and elsewhere: I recommend restraint in the use of “hitherto”

Agreed. We change this word by “until now” in order to restraint the use of “hitherto”.

Line 84: what does “deep-sea” mean here?

Yes, we change the word deep-sea by bathypelagic zone which is the right word in this phrase. The reviewer should understand that it is difficult to avoid repetition of some words along the manuscript.

Line 98 onwards: I found some of this paragraph to be repetitive. Also, the insertion of the regression relationships into the text with no definition of LnPP, LnB etc does not provide the reader with any useful information. Either the relationships need to be better contextualized/explained/defined, or the regression statistics need to be presented differently.

Agreed. The reviewer is right as net primary production is stated as NPP along the text and in the relationships is stated PP instead of NPP. Similarly, zooplankton biomass should be written as ZB instead of simply B in the equation. This is now changed in the text.

Line 124 onwards: see my comment above about the N isotope data.

This is now changed in lines 126-127 in the new version of the manuscript as explained above.

Line 142: what observation?

Agreed. We now refer to Supplementary Figure 3 in the previous phrase in order to support the word “observation”.

Line 144: the authors have a myriad of ways of referring to the deep ocean, some more specific than others. Here they now use “twilight zone”. It’s confusing to the reader to have so many different terms used to refer to the same thing with varying degrees of specificity. I recommend being more selective, as also as precise as possible each time – when you say deep sea, for instance, do you mean the entire deep ocean, certain layers, etc.

As stated above, it is difficult to avoid repetition of some words along the manuscript. Normally, we tried to use “deep-sea” or “dark ocean” referring to the bathypelagic zone and it is used when we name the other layers, the epi- and mesopelagic zones (as it was the case above). The word “twilight zone” is normally used to name the mesopelagic zone as there is some match between both layers. It is only a way to avoid repetition of the same words along the manuscript, something complicated to non-English native speakers. We decided to keep the words “twilight zone” and “deep sea” because it is rather clear that we are referring to layers below the epipelagic zone.

Line 147: from what? How?

We changed the word “being” here as perhaps is more appropriate here than “with”.

Line 149: there are aspects of this paragraph that are repetitive. I suggest revisiting for clarity and repetition. Likewise, line 166-168 and 173-175.

Agreed. The reviewer is right. We were quite repetitive with this concept. Thus, we deleted the phrase "...provide a minimum estimate of carbon demands, as it..." in line 165, and also the same concept in line 169 deleting the word "minimum".

Line 195: that can be supported by the passive flux? Also, please see my comment above about DOC. Finally, "an important contribution from active carbon flux" is vague – be specific in what you mean by "important".

Agreed. Done. We also changed the word "important" by "additional" which is the true sense of the phrase (line 204). The comment about DOC is explained above.

Line 206: I don't understand this sentence – "fragmentation" (this is the first use of this term), "be the rule" (what is meant by this?)

The reviewer is right. We refer to particle fragmentation because of, among others, (small?) zooplankton processing of particles tend to decrease their size and decrease passive flux. So, we include here the word "particles" (line 215).

Line 223: if there have been 50 years of research efforts, a few citations are required here.

Sorry, we do not understand this requirement as there is an important information about zooplankton vertical distribution during the last decades but almost no effort to compare primary production and zooplankton biomass in deep waters. So, we consider unfruitful to show any citation here as to say these authors did not find such a relationship is not polite.

Line 229: surface chlorophyll?

The reviewer is right. We added the word "surface" now in line 238.

Line 234: horizontal transport of what?

Agreed. We added "water mass" in line 243.

Line 245-248: please clarify what you mean here.

We have rewritten the phrase in lines 254-256 in order to explain better the meaning.

Line 258: please see my comment above about the implications of all this for the biological pump.

As stated above, we explain this in lines 262-271.

Methods

Line 418: "published conversion factors" – how much error might this introduce.

We think this manuscript is not the right place to discuss this which in fact is an important issue in many studies about biomass in the ocean performed by many

different authors. In any case, the percentage of error in this exercise is much lower than the zooplankton biomass variability in the ocean (2-3 orders of magnitude). This is an important problem but there is no other way than the use of conversion factors.

Line 455: protein to dry weight conversion – should there be a reference for this? If not, where did the number come from?

The reviewer is right. We added here the reference where this ratio was published (line 529)..

I think a little more detail needs to be provided on the Red Sea data that were excluded.

We think this is explained as we observed very high NPP and low biomass promoting a clear outlier. We exactly do not know the cause of the high NPP signal but it should be related to problems of the remote sensing technology in this small sea.

Data availability: I am not aware of the requirements of the journal, but surely the data should be made available in a public repository – the days of “contact the first author” have passed and it’s important for the integrity of the work (as per the FAIR data principles), that the data be publicly available. I encourage the authors to take this seriously.

The reviewer is right. We send the data to PANGEA repository and we are now waiting for the DOI. The reference will appear in the text once it will be accepted and before the printing proofs.

Reviewer #3 (Remarks to the Author):

The authors have done a good job with revising the ms and the additional figures help to better understand the processes. I have only a few additional comments.

L 87: I would suggest that this paragraph includes a short description of the approach taken, like N-Isotopes and respiration, particularly since the methods are given at the end and a big part of the results are moved to the supplementary information. This will help the reader to better follow the results and conclusions.

Agreed. We added the isotope and respiration information at the end of the paragraph (lines 93-95).

L 150: change to 'than previously estimated'. Can you include a reference supporting this statement?

Agreed. We also added the references supporting the statement (line 152).

L 233: should this be ‘..., such as the pelagic zone’?

Agreed. Done (line 242).

Figure 1: In this figure ‘oceanic biome’ is the main grouping variable but the biomes have not been described/defined before. A supplementary map showing the regions could be a solution to this.

Agreed. We added in Figure 1 the reference to oceanic biomes as a better indication to this.

Figure 2: The figure legend states, ‘Note that both axes are in logarithmic scale.’ But the x-axis is log with base 2, while the y-axis is log with base 10, even though annotated as if it was log with base 2. Perhaps this can be clarified in the figure legend to avoid confusion or misinterpretation.

Both axis in Figure 2 are in same base. Although the numbers in the x-axis give other impression, see the tickmarks. We increased the size of the tickmarks for clarity.

Figure 3: Here, alignment of the map with the graph would give a clearer interpretation, and maybe including the oceanic biomes.

We think this alignment would blur the graph as for instance the start of the section is in an eastern position. We assume the reader know the places (Brasil, South Africa,...). We also consider to add the biomes to complicate in excess the Figure.

Figure 4: Explain what the difference between green and red arrows is to avoid misinterpretation

Agreed. We added the color indication of arrows (green, orange, red) for the passive, DOC, and active flux in the Figure legend.

Figure 5: Clarify what the difference between red and blue migration patterns is.

We deleted this Figure as suggested by reviewer #1.

Supplementary Table 1: Since focus in this table is put on Biome types, it would be interesting to see the statistical significance of this term.

The statistical significance is given in column 4 of the Table.

Reviewers' Comments:

Reviewer #2:

Remarks to the Author:

This is a novel and data-rich study that I think will be of great interest to community. In my previous review I raised two major points (the role of DOC and the implication of the findings discussed herein for the biological pump) that I now feel have been adequately addressed by the authors.

In re-reading the manuscript, I found that the authors switched tense a number of times, so I suggest that they try to make this more consistent. There is also some repetition in word choice in a few of the sentences (e.g., "assumed" in line 67 and 68). These comments are entirely superficial, however, and do not detract from the quality of the manuscript.

A few minor suggestions:

Line 66: do you mean "exponentially decreasing nature..."?

Line 77-78: I wasn't really sure what the authors were trying to say for point (3).

Line 94: nitrogen stable isotopic composition of ??

Line 132: in place of "those" I think you mean zooplankton?

Line 133: Whose dependence?

Line 149: I suggest replacing "being" with "with".

Line 190 and 191: I didn't understand the clause "where it supports an estimated DOC...." I think something is missing from the sentence?

Line 221: "higher" nitrogen isotopic composition, not heavier nitrogen isotopic composition.

Reviewer #2 comments and Editorial request:

Reviewer #2 (Remarks to the Author):

This is a novel and data-rich study that I think will be of great interest to community. In my previous review I raised two major points (the role of DOC and the implication of the findings discussed herein for the biological pump) that I now feel have been adequately addressed by the authors.

Agreed.

In re-reading the manuscript, I found that the authors switched tense a number of times, so I suggest that they try to make this more consistent. There is also some repetition in word choice in a few of the sentences (e.g., "assumed" in line 67 and 68). These comments are entirely superficial, however, and do not detract from the quality of the manuscript.

A few minor suggestions:

Line 66: do you mean "exponentially decreasing nature..."?

Done. Agreed (Page 3, paragraph 1, line 7).

Line 77-78: I wasn't really sure what the authors were trying to say for point (3).

We added the word ...often... in order to clarify. (Page 3, paragraph 2, line 10).

Line 94: nitrogen stable isotopic composition of ??
...of zooplankton... Agreed (Page 4, paragraph 2, line 9).

Line 132: in place of "those" I think you mean zooplankton?

In this case we mean "provinces", so we added ...in provinces of the Coastal biome,... (Page 6, line 12).

Line 133: Whose dependence?

We changed the whole sentence : "This enrichment was significant in the Trades biome provinces but not in provinces of the Coastal biome, thus suggesting the existence of regional differences in the links between zooplankton and surface productivity." (Page 6, lines 11-13).

Line 149: I suggest replacing "being" with "with".

Done (Page 7, line 4).

Line 190 and 191: I didn't understand the clause "where it supports an estimated DOC...." I think something is missing from the sentence?

We refer to DOC supporting only 8.4% of respiratory demands in the bathypelagic layer. We think it is clear so we did not changed the phrase. We could add "in the bathypelagic layer" after demands but it seems a redundancy (Page 8, paragraph 3, line 5).

Line 221: "higher" nitrogen isotopic composition, not heavier nitrogen isotopic composition.

Done. Agreed (Page 9, paragraph 2, line 15).